# Study of Raw Material Pretreatment and the Microbiota Selection Effect on the Composting Process Efficiency

Abdo Tannouri [1,2,*], Ziad Rizk [1], Marina Daccache [3], Chantal Ghanem [1], Valérie Azzi [1], Richard G. Maroun [2], Zeina Hobaika [2] and Dominique Salameh [2]

1 Lebanese Agriculture Research Institute (LARI), Zahle P.O. Box 287, Lebanon; ziadrizk_ce@hotmail.com (Z.R.); chantalghanem1@hotmail.com (C.G.); valerie.azzi@hotmail.com (V.A.)
2 Faculté des Sciences, Campus des Sciences et Technologies, Université Saint Joseph Beyrouth, Beirut P.O. Box 17-5208, Lebanon; richard.maroun@usj.edu.lb (R.G.M.); zeina.hobaika@usj.edu.lb (Z.H.); dominique.salameh@usj.edu.lb (D.S.)
3 Ecole Supérieure d'Ingénieurs de Beyrouth (ESIB), Saint-Joseph University, Beirut P.O. Box 11-514, Lebanon; marina.daccache1@usj.edu.lb
* Correspondence: abdotannoury@hotmail.com

**Abstract:** Lignocellulosic is a carbon source biomass composed of cellulose, hemicelluloses, and lignin, which are strongly associated with each other. This fact makes them hardly degradable by produced microbial enzymes when introduced to compost piles. To address this problem, a primary single or combined pretreatment method of this biomass allows for the separation of these complex, interlinked fractions, allowing a better accessibility for microbial enzymes. However, the rugged lignin component, in addition to several produced by products from these pretreatments, inhibit the microbial activity. For this, the optimization of these treatments with other interfering parameters is the base for a successful composting process. In this work, nine compost piles were initiated, in which their lignocellulosic fraction was subjected to chemical and microbiological treatments alone or combined while preserving a control. The obtained results showed that the combined pre-treatment of the primary organic raw materials with 10% NaOH and adapted microbial inoculum at 2.5% was the best suited for compost piles in Mediterranean regions. This treatment ensured the quickening of the composting process by 15 days, while yielding a final compost of a higher quality in regard to its physic-chemical characteristics, especially its C:N and CC values. Furthermore, it ensured a higher sanitation through the elimination of different microbial pathogens from the final compost, by means of the secondary metabolites produced by the microbial adapted consortia. This 'tailor-made' process could be replicated for the treatment of other generated sources of organic raw materials within the Mediterranean region.

**Keywords:** chemical pretreatment; lignocellulosic biomass; inoculum



## 1. Introduction

Composting is a process involving a complex ecosystem with many interacting factors, including several physicochemical, biochemical, and microbial community changes. The growth of the microbiome that populates a composting system is temperature dependent. Therefore, three classes of microbiome can be distinguished depending on the growing temperature as follows: psychrophilic (temperature range between 0 and 25 °C), mesophilic (active between 25 and 40 °C), and thermophilic (temperature range between 50 °C and 65 °C) [1]. This microbiome degrades unstable organic substrates into more stable, humified forms and inorganic products that generate heat and water as metabolic waste [2].

In this context, the decomposing microbial population varies during the composting process depending on the temperature fluctuations. Bacteria are usually present in large numbers throughout the whole composting period and are a major contributor to degradation processes, as they are responsible for 80 to 90% of the microbial activity [3]. The



most common cellulolytic fungi species observed in composting materials are *Aspergillus*, *Penicillin*, *Fusarium*, *Trichoderma*, and *Chaetomonium* [4]. An effective process optimization relies on the destruction of the pathogens present in the used feedstock, taking into account the relationship between the temperature and the time for pathogen kill. In fact, a high temperature for a short period of time may be just as effective as a lower temperature for a longer duration [5]. Successful composting depends on several factors, which have both direct and indirect influences on the activities of microorganisms, including the type of raw material, its nutrient composition, and its physical characteristics [6]. Particle size and moisture content are both critical criteria for optimal composting. Particle size affects not only moisture retention, but also the free air space and porosity of the compost mixture. A minimum moisture content between 50 and 60% is most desirable for good microbial activity [7].

The level of biodegradability of organic matter can vary according to its quality and quantity [8]. Generally, the level of organic matter in compost is lower than that present in the primary raw materials and it may vary between 20 and 60% [9]. Therefore, carbon and nitrogen are essential for the composting process, to provide the main source of energy and boost the growth of the microbial population [10]. Usually, organic carbon includes two fractions, the Total Organic Carbon (TOC) and the mineral carbon, present in the form of carbonates and bicarbonates [11]. As for nitrogen, it usually represents 1 to 4% of the total dry matter in the compost, of which mineral nitrogen represents at least 10% [12]. At the end of the composting, organic matter is mineralized, especially into nitrate ($NO_3^-$), of which a first part is reincorporated into the microbial metabolism, a second part is incorporated in the humified organic matter, and a third part is freely present in the compost [13].

During the early stages of composting, the pH decreases due to the production of organic acid, especially when the composting involves agricultural wastes [14], but as the process continues, these acids are converted into $CH_4$ and $CO_2$, turning the pH into neutral. Researchers have found that a mature compost should have a pH between 7 and 9 [15].

Maturity and stability are important aspects of composting, since they are related to its application in the field, due to its organic matter and nutrients content [16]. Several maturity indices have been adopted by different researchers, for example, Cation Exchange Capacity (CEC), which should be calculated per unit ash [17], with a C:N ratio between 25 and 30 to obtain good-quality compost [18]. Studies differ in terms of the C:N ratio; some researchers have recommended ratios between 10 and 13 [19] and others have recommended a C:N of 18 [20].

To favor the decomposition process, many investors have relied on the pretreatment of the lignocellulosic fraction with physical, microbiological, thermal, or chemical methods. This pretreatment has been used to disrupt the close inter-component association between the main constituents of the plant cell wall [21] and remove the barriers that make the native biomass recalcitrant through the solubilization of hemicelluloses and/or lignin, which coat the cellulose, make it easily amenable to enzymatic degradation, and increase the levels and yield of reducing sugars [22]. For this, a detailed understanding of the structure of the polysaccharides forming the lignocellulosic fraction is vital to understanding the obstacles limiting their composting. The lignocellulosic fraction consists mainly of three types: cellulose (linear homopolysaccharide polymer), hemicelluloses (heteropolysaccharides with a low degree of polymerization that bind the cellulose and the lignin), and lignin (a complex aromatic polymer) bounded to each other [23]. Hemicelluloses and lignin are normally dissolved in water at 180 °C under neutral conditions [24]. Their solubilization under acidic, neutral, or alkaline conditions is dependent on precursors such as p-coumaryl, coniferyl, sinapyl alcohol, or their binding structures [25]. To add, wood extractives are a heterogeneous group of compounds that are not present in the lignocellulosic fraction, but appear as a result of chemical processes and are mainly composed of terpenes, fats, waxes, and phenolic compounds. It is important to note that high levels of polyphenols in the mixture may partially or totally inhibit the bacterial growth [26].

Chemical pretreatment could be fulfilled by acidic or alkaline reagents and each method has its advantages and disadvantages. Acidic pretreatment with HCl, $H_2SO_4$, $H_3PO_4$, or $HNO_3$ has a double advantage, as most of the hydrolytic microorganisms can withstand acidic conditions and alter the rigid structure of biomass [27]. During this pretreatment, hemicelluloses and amorphous parts of the cellulosic fraction are hydrolyzed, based on the concentration level of the used acid, while lignin is condensed and precipitates [28]. It should be noted that a pretreatment with high acid concentrations may result in the production of certain inhibitory by-products, such as furfural and hydroxymethyl-furfural, and therefore, the use of diluted acid is usually recommended [29]. Other disadvantages of this pretreatment include the loss of fermentable sugars due to the excessive degradation of complex substrates, the high cost of the acidic products used, and the additional cost required to neutralize the substrate after their use and before the start of the composting process [30].

Alkaline reagents gather derivatives of sodium hydroxyls, of which sodium hydroxide (NaOH) is considered to be the most efficient, in addition to potassium, calcium, and ammonium salts [31]. When applied to the lignocellulosic fraction, alkaline reagents target the lateral chains of esters and glucosides, causing structural modifications in the lignin, in addition to the swelling and recrystallization of cellulose [32]. Pretreatment with mild alkaline reagents is easier compared to acidic reagents, and could be easily performed at ambient and high temperatures in case it must be kept for longer times. In addition, a neutralization step is recommended for the elimination of inhibitory by-products and lignin [33]. Alkaline pretreatment is considered advantageous, as the cost of the used chemicals is relatively low. However, its only disadvantage is that the process requires larger amounts of water and the presence of large quantities of salts, which are eliminated using a specific procedure, making the cost higher [34]. To note also, that extrusion is another used pretreatment method applied to reduce the residual size of the lignocellulosic biomass, allowing a better accessibility to microbial enzymes. According to [35], due to the varying types of lignocellulosic biomass, the feasibility and economic analysis of extrusion pretreatment are key factors for the success of pretreatment technologies.

On the other hand, the biological process is considered efficient, as it requires less energy at a low cost and is more environmentally friendly. Worth noting is that the biological process is relatively slow, since it relies on the existing microbiota on the primary organic raw materials. In most cases, these microbiota are not active in the decomposing process, and/or have low antagonistic potential regarding their secondary metabolites. Thus, they are subjected to severe competition by existing pathogens such as *Listeria monocytogenes* and *Salmonella Typhimurium* [36]. For this, the selection and application of an abundant number of adapted microorganisms, especially *Actinobacteria* and fungus, existing naturally in the environment, which can be easily invested and used in the pretreatment of biomass, targeting cellulose and hemicelluloses [37], helps to fasten the process. Decomposition by fungi occurs through two types of extracellular system: the production of hydrolase for the degradation of polysaccharide, and the oxidative system for the degradation of lignin and extension of phenyl rings [38]. On the contrary, *Actinobacteria* are primarily saprophytes and contribute significantly to the turnover of complex biopolymers, such as lignocellulose, hemicelluloses, pectin, keratin, and chitin.

In previous work [36], eleven bacterial isolates, including three different strains of *Bacillus subtilis* (CBI 1, CBI 2, and CBI 7), two different strains of *Providencia* sp. (CBI 3 and CBI 11), *Alcaligens* sp. (CBI 4), *Pseudomonas* sp. *20_BN* (CBI 5), *Bacillus pseudomycoides* (CBI 6), *Arthrobacter* sp.(CBI 8), *Myroides* sp.(CBI 9), and *Pseudomonas* sp. (CBI 10), were selected from Lebanese compost piles and characterized biochemically and genetically; the antagonistic potential of their secondary metabolites against several pathogens were validated (bacterial isolates 1, 5, and 9 inhibited the growth of both *Salmonella typhimurium* and *Escherichia coli*, bacterial isolate 6 acted against *Salmonella typhimurium*, and those of bacterial isolate 8 had inhibitory activity against *Listeria monocytogenes)*, and against fungal pathogens (secondary metabolites of all the bacterial isolates were able to inhibit *Fusarium*

*oxysporum*, and those of bacterial isolate 7 were able to inhibit *Alternaria solani)* that might be present in the primary organic matter used in compost piles. A second set of tests covered biochemical and enzymatic tests for both profiling issues, and understanding their decomposition potential.

In this context, most existing composting units worldwide rely on the existing microbiota in the primary raw materials instead of a specific adapted inoculum of microbiota capable of initiating and maintaining a robust decomposition process. Moreover, most collected comingled wastes are not clean, since they are mixed with physical impurities (metal, glass, or non-recyclables) or polluted by chemical residues (pesticides, antibiotics, or heavy metals...) that strongly affect the persistence and activity of the present microbiota.

The aim of this work is to develop a robust inoculum containing these adapted strains and study their ability to launch and maintain a strong, more efficient decomposition process, with more sanitation potential through their secondary metabolites within compost piles. To have a clear understanding of the different interaction factors, field tests covered the usage of this inoculum at different concentrations alone or in accordance with the treatment of the lignocellulosic fraction with acidic (HCl 10%) or alkaline (NaOH 10%) chemical solutions, in comparison to a control.

## 2. Materials and Methods

### 2.1. Field Trials

The use of raw materials consisted of cow manure and a mix of wood remnants (vines, fruit, and forest trees) chopped to a size from 3 to 5 cm. The field trials were conducted in Bkaatouta, Keserwan district, Mount Lebanon, while adopting the windrow system. Nine compost piles of 130 Kg each were constructed, containing 70% manure (90 Kg) and 30% wood remnants (40 Kg). The piles were subjected to various treatments (microbial, acidic, or alkaline [39], or both acidic and alkaline treatments), which are detailed in Table 1.

**Table 1.** Treatment(s) used for each pile, +: treatment used; −: treatment not used.

| | Treatments | | | |
|---|---|---|---|---|
| **Pile Number** | **NaOH 10% (0.1 M)** | **HCl 10% (3.2 M)** | **Inoculum 1% [36]** | **Inoculum 2.5% [36]** |
| Control | − | − | − | − |
| Pile 1 | + | − | + | − |
| Pile 2 | + | − | − | + |
| Pile 3 | + | − | − | − |
| Pile 4 | − | − | + | − |
| Pile 5 | − | − | − | + |
| Pile 6 | − | + | + | − |
| Pile 7 | − | + | − | + |
| Pile 8 | − | + | − | − |

The microbial treatment was prepared with 11 strains belonging to the authors' laboratory strains collection [40], previously isolated and identified from a spontaneous fermentation of Lebanese compost piles, and these were incubated on Luria Bertani broth Sigma-Aldrich (Munich, Germany) for 48 h at 30 °C to reach a cell concentration of $1 \times 10^8$ CFU.mL$^{-1}$ [41]. Then, they were mixed at the field level before their application to the piles. A pile was not subjected to any treatment and was considered as a control. The 11 selected Compost Bacterial Isolates (CBI) were isolated on International Streptomyces Project-2 Medium (ISP2) [42] and subjected to biochemical testing, as recommended by the International Streptomyces project for mycelial organisms (Table 2).

**Table 2.** Characteristics of the wood remnants, the cow manure used, and the Compost Bacterial Isolates (CBI).

| Shredded Wood Remnants | | | | | | | |
|---|---|---|---|---|---|---|---|
| Timing | Lignin | Hemicellulose | Cellulose | | | | |
| Pile Initiation | 46 | 17 | 37 | | | | |
| End of composting | 19 | 6 | 25 | | | | |
| **Cow manure profiling** | | | | | | | |
| Organic Matter (OM %) | Nitrogen % | $P_2O_5$ % | $K_2O$ % | pH | EC | C % | C:N |
| 86.06 | 1.73 | 0.98 | 0.73 | 7.2 | 0.65 | 44.46 | 25.7 |
| **CBI \*** | **Nutritional profile and growth characteristics** | | | | | | |
| CBI 1 | Sugars, amino acids, and starch (Thermophile T° range 40–55 °C), secondary metabolites inhibited: *Salmonella typhimurium* and *Escherichia coli* | | | | | | |
| CBI 2 | Amino acids, organic compounds, and starch (Mesophile T° range 30–45 °C) | | | | | | |
| CBI 3 | Organic compounds and proteins (casein) (Thermophile T° range 40–50 °C) | | | | | | |
| CBI 4 | Amino acids (serine), organic compounds, and proteins (Mesophile T° range 30–45 °C) | | | | | | |
| CBI 5 | Sugars, proteins, amino acids, organic compounds, and starch (Mesophilic T° range 37–45 °C), secondary metabolites inhibited: *Salmonella typhimurium* and *Escherichia coli* | | | | | | |
| CBI 6 | Sugars (melezitose), amino acids (proline and serine), proteins (casein), and starch (Thermophile T° range 40–55 °C), secondary metabolites inhibited: *Salmonella typhimurium* | | | | | | |
| CBI 7 | Amino acids, starch, proteins, and organic compounds (Mesophile T° range 37–45 °C) | | | | | | |
| CBI 8 | Sugars, amino acids, starch, organic compounds, and proteins (Mesophile T° range 37–45 °C) | | | | | | |
| CBI 9 | Sugars, amino acids, organic compounds, and starch (Thermophile T° range 40–50 °C), secondary metabolites inhibited: *Salmonella typhimurium* and *Escherichia coli* | | | | | | |
| CBI 10 | Amino acids, organic compounds, starch, and proteins (Mesophile T° range 30–45 °C) | | | | | | |
| CBI 11 | Amino acids, starch, organic compounds, and proteins (Thermophile T° range 40–50 °C) | | | | | | |

\* CBI Compost Bacterial Isolates.

### 2.2. Temperature and Humidity Monitoring

The temperature was monitored on daily basis using a probe thermometer (dial compost thermometer, Rapitest). The humidity was kept at the field level between 50 and 60% using the "hand-squeeze method" or a sponge test developed by Will Bakx of Sonoma Compost [43].

### 2.3. Sampling and Testing

Samplings were conducted seven times; at T0, representing the launching date of the compost piles, and successively at T1 (day 15/360 h), at T2 (day 30/720 h), at T3 (day 45/1080 h), at T4 (day 60/1440 h), at T5 (day 75/1800 h), and at T6 (day 90/2160 h) from launching the composting process. These recuperated samples were analyzed for several features. The colony count of the mycelial bacterial organisms was conducted on ISP2 medium, as recommended by the International Streptomyces project for mycelial organisms. Physico-chemical characterizations, namely the pH [44], conductivity [45], total nitrogen [46], C:N ratio [47], dry matter [48], volatile matter, organic carbon [49], organic matter [39], phosphorus [50], and potassium [51], were performed.

The biochemical fractioning of the lignocellulosic biomass was estimated for the 9 piles at the 7 different times using Van-Soest's method, supplying used chemicals from Sigma-Aldrich (Munich, Germany) [39] as follows: the total soluble compounds were extracted with a neutral detergent solution ($N_{a2}HPO_4$, sodium tetraborate, α-amylase, sodium EDTA, sodium lauryl sulfate, and sodium sulfite), acting on the sample for 1 h at 100 °C. Then, hemicelluloses were extracted using an acid detergent solution (cetyltrimethylammonium bromide 20 kg m$^{-3}$, 98 kg m$^{-3}$ of $H_2SO_4$), acting for 1 h at 100 °C. All the extracted fractions were separated from the used neutral and acidic detergents via filtration. Cellulose was extracted after treatment with 1317 kg m$^{-3}$ of $H_2SO_4$ for 3 h at ambient temperature (20 ± 1 °C). The filtration residue corresponded to the lignin associated with the inorganic material. The lignin fraction was thus determined in a muffle furnace after 4 h at 550 °C.

The hemicellulose and cellulose content were calculated as the difference between the tested fractions, respectively, as shown in the below formulas:

$$\text{Hemicelluloses content} = \text{Neutral Detergent Fiber (NDF)} - \text{Acid Detergent Fiber (ADF)} \tag{1}$$

$$\text{Cellulose content} = \text{Acid Detergent Fiber (ADF)} - \text{Acid Detergent Lignin (ADL)} \tag{2}$$

### 2.4. Final Compost Quality Testing

In addition to the mentioned analyses in the previous section, *Salmonella* spp. [52], *Escherichia coli* [53], *Staphylococcus aureus* [54], *Bacillus* spp. [55], *Enterococci* spp. [56], and mercury (Hg) [57] were applied to the nine piles at day 90 (T6).

All the culture media components and chemicals were supplied by Sigma-Aldrich (Munich-Germany), except for the Yeast Extract and the peptone, which were supplied by Oxoid (Hampshire-England).

### 2.5. Statistical Analysis

All the data reported represent an average of three replicates with standard deviation. XL-STAT SPSS software (Version 2014.5.03) was used to treat the data. The observations and correlations between all the parameters were tested using a principal component analysis (PCA) and Pearson (n) type. Two-way ANOVA (generalized linear models) at $p < 0.05$ and Tukey's multiple range test ($\alpha = 5\%$) were used to assess the statistical significance differences of the colony count and C:N ratio of the 9 piles.

## 3. Results and Discussion

### 3.1. Temperature Variation

The temperature patterns clearly showed two main phases during the composting process, a fermentation/oxidative phase and a maturation phase (Figure 1a). The control showed the same temperature patterns as described in the literature [58]. In the fermentation phase, a rapid increase in the temperature was noted in all the piles during the first 48 h. This increase in the temperature may have been due to the degradation of the present simple sugars and amino acids in the labile organic matter. After 48 h, a thermophilic phase was observed, in which a high rate of biodegradation and the mineralization of more complex organic materials took place by other types of existing microbiota [2]. A fluctuation in the thermophilic phase was noted at T2 for the different piles. This fluctuation was due to the various chemical and microbial treatments applied and to the heat exchange with the atmosphere, thereby affecting the synergy between the existing microbiota [59]. In fact, a synergy existed between the microbiota during composting [3]. Some of the simple molecules produced by the degradation process would be used by other microbiota throughout the process. Studies have shown that, at temperatures above 70 °C, only the enzymes produced during the early phases will contribute to the degradation process [60]. In this study, such values were not achieved due to the economy of scale, since the size of the piles was only 130 Kg. A sequential decrease in the temperature was detected in all the piles, with distinct readings at T4, depending on the applied treatment. The oxidation phase was followed by a maturation phase, in which the activity of the microbiota decreased due to the depletion of the organic matter and the piles cooled to reach ambient temperature. During this phase, the secondary reactions of condensation and polymerization predominated, leading to a stable formation of humus and humic acid from the lignin, polysaccharides, and nitrogen compounds [61]. The detected ambient temperature in the composting piles was between 27 and 30 °C (Figure 1a).

A significant temperature increase was observed for piles 2 and 3, reaching, respectively, 61 °C and 60 °C after 48 h, in comparison to the control (Figure 1a). This high temperature increase may have been associated with the alkaline treatment (NaOH 10%). In fact, it allowed for the solubilization of the existing hemicelluloses and part of the lignin,

thus facilitating their degradation by microbial enzymes and accelerating the oxidation phase and the reduction in its timing. The addition of the specific microbial inoculum (2.5%) to pile 2 induced greater microbial activity that could compete with other existing pathogens for substrate consumption and space. In addition, the microbial inoculum secreted antagonistic secondary metabolites in the medium that inhibited the growth of other existing species. The activities of NaOH (10%) and the microbial inoculum ensured synchronized degradation activity within pile 2 through both the oxidative and maturation phases, guaranteeing the availability of the needed, easily degradable source of nutrition required for continued intensive microbial activity. Studying the impact of this treatment on the other covered parameters will help to validate this statement.

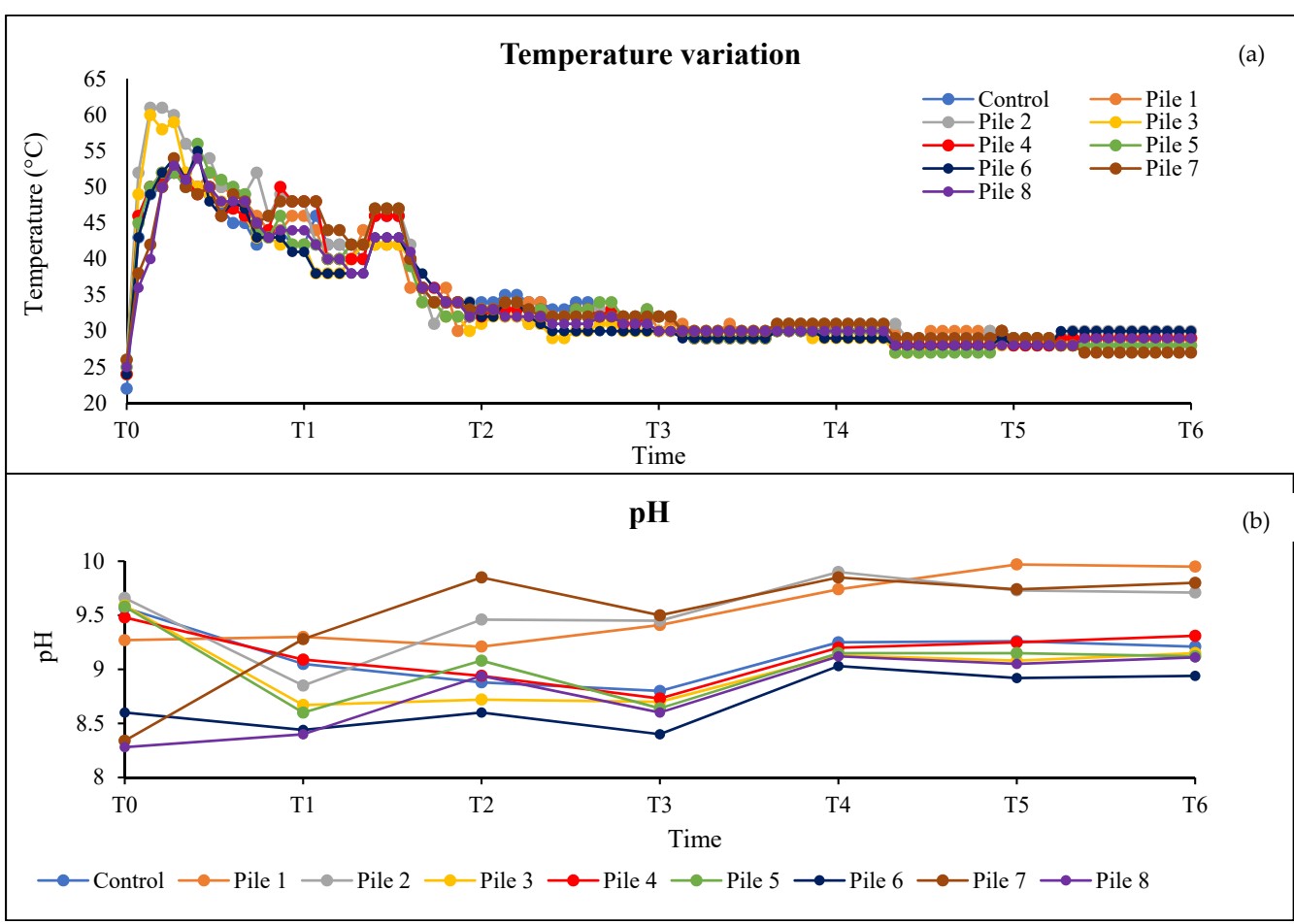

**Figure 1.** Variation of: (**a**) temperature, (**b**) pH within the 9 compost piles.

Same patterns were obtained for piles 4 and 5, where the temperatures were higher than the control due to the higher microbial activity supplemented through these inoculums, which allowed for more competition and higher degradation potentials. Piles 1, 4, and 5 showed a slower evolution of the temperature, where the temperature reached a maximum of 50 °C (Figure 1a) in comparison to piles 2 and 3 on the 3rd day of the piles' initiation.

The temperature variation in piles 6, 7, and 8 showed similar patterns of evolution, as the control reached a maximum of 50 °C on the 4th day from the initiation date. A difference within these piles was the similarity to other piles regarding the rise in the temperature during the first 24 h, which was delayed during the following 24 h in comparison to the other piles. This fact could have been due to the production of inhibitory compounds, such as furfural, following the acidic treatment of the carbon fraction, which could have altered the microbial activity, causing a decrease in the pile temperature. After 48 h, the

concentration of these inhibitory compounds was decreased by climatic and management practices, allowing the microbial activity to be launched again.

*3.2. pH Variation*

The pH values in piles 1, 2, and 3 at T0 were 9.27, 9.66, and 9.59, respectively, and for piles 4 and 5, the pH values were 9.48 and 9.58, respectively. As for the control, the registered pH was 9.57. At T1, the pH in piles 2 and 3 had a similar trend and showed a fast decrease in the pH by 1.66% and 3.66%, respectively (Figure 1b). This fast decrease was due to the solubilization of the hemicelluloses and part of the lignin by the NaOH solution, coupled with the loss of ammonia due to the volatilization under NaOH concentrations higher than 6% [62]. The variation between the two piles could also have been due to the treatment with the 2.5% microbial inoculum of pile 2 affecting the quantities of the produced and accumulated organic compounds, such as acetone, butanol, propanol, lactate, and butyrate, etc. [63], generated usually during the first hydrolysis phase of the composting. In pile 4, the pH signaled a decrease by 4.11% (Figure 1b). For the same time duration, the pH patterns for the control and pile 4 showed a progressive decrease until T3 by 8.05% and 7.91%, respectively (Figure 1b), due to the high competition with pathogens in the control pile, since the existing microbiota were not adapted and a low microbial inoculum concentration was applied to pile 4 (1%), in addition to the bad impact of the produced volatile organic compounds (VOCs) from yards waste decomposition under aerobic conditions [64]. The pH within pile 1 showed a slight increase from T0 until T1, which could have been due to the improper rinsing of the treated lignocellulosic fraction following its dipping in the NaOH solution, causing the integration of the liberated base within the treated raw materials, in addition to ammonia production from the degradation of amines (proteins and amino acids).

The pH increased rapidly from T0 until T2 in piles 2 and 5 and progressively in pile 3, reaching, respectively, 9.46, 9.08, and 8.72 (Figure 1b). This increase was due to the production of ammonia following the degradation of amines, and the liberation of the integrated bases previously within the organic matter. This fact showed the degradation potential of the used bacterial inoculum applied to our piles, since the nutritional profiling showed that all of them were active in the degradation of proteins from different sources (ex: CBI 3 and CBI 6 degrade casein), amino acids, and sugars (Table 2).

As for the last three piles, piles 6, 7, and 8, their initial pHs at T0 were, respectively, 8.6, 8.34, and 8.28 (Figure 1b). These low values could have been due to the persistence of HCl residues within the treated lignocellulosic fraction, which required a prolonged time of rinsing. Other reasons include the organic acid production following the hydrolysis of the hemicelluloses and amorphous part of the cellulose, the condensation and precipitation of the lignin [28], and the degradation process by the existing microbiota. Even though the existing microbiota could withstand acidic treatment by itself, several inhibitory by-products, such as furfural and hydroxymethyl-furfural, were produced within the used organic materials [29], which could limit the growth of the microbial activity. This was clearly seen during the first 15 days within pile 7, which received a higher concentration of the bacterial inoculum (2.5%), allowing for a more prominent decomposition activity of the existing organic matter than that in pile 6, which received a lower concentration (1%) of the bacterial inoculum, and that of pile 8, which did not receive any microbial treatment. Once the limiting factors were eliminated due to climatic conditions and the pile management (pile humidification), in addition to chemical reactions, the microbial activity was re-launched. A fluctuation in the pH was noted among the different piles with a varying intensity between T2 and T5, with an exception in pile 1, which preserved a constant increase during the whole period and expanded until T6. Instead, the applied treatments to pile 2 allowed an easier accessibility for microbial enzymes, and a faster decomposition was noted. After T5, the pHs of the different piles became stable due to the decrease in the available nutrients and by that, the activity of the microbial populations. Worth noting is that the basic pH is an indicator of compost stability [15], and the primary

and final pH may vary according to the type of organic matter used in the primary mixture. The pH values in urban wastes range between 5 and 9, and acidity is not a limiting factor in composting [65].

The obtained results for both the temperature and pH variations had the same behavior, as stated by both [18,63], regarding the variations developed from different applied chemical and microbial treatments and the identification of the oxidative phase within the first 55 to 60 days from launching the composting process, including the mesophilic phase within the first 48 h, the thermophilic phase during the remaining period, and the maturation phase, which took place for the last 30 to 35 days.

### 3.3. Principal Component Data Analysis (PCA) of All the Parameters

The results of the PCA showed that the eigenvalues of the three components explained 62.02% of the total variance correlation (Figure 2).

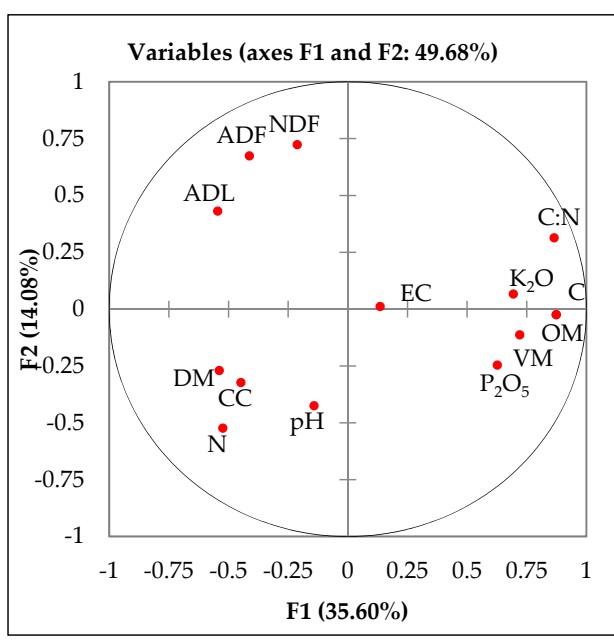
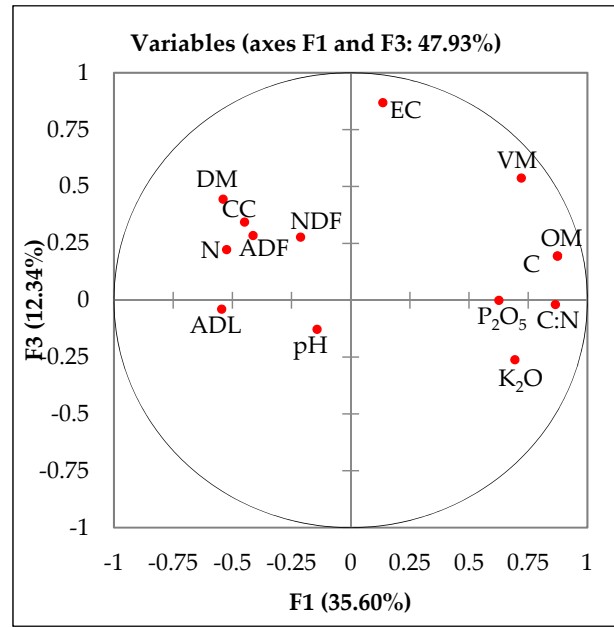

**Figure 2.** Pearson correlation of all the parameters analyzed during the composting process in the 9 piles. ADF: Acid Detergent Fiber; NDF: Neutral Detergent Fiber; ADL: Acid Detergent Lignin; DM: Dry Matter; CC: Colony Count; VM: Volatile Matter; OM: Organic Matter; and EC: Electrical Conductivity.

The ADF was positively correlated with the ADL and NDF (*R* between 0.4669 and 0.6410), which is in agreement with what was described previously by the pH and temperature variation (Table 3). The difference between the NDF and ADF fractions allows for the identification of the hemicellulose content, and the difference between the ADF and ADL allows for the identification of the cellulose content, as stated by [38].

The Colony Count (CC) was positively correlated with the nitrogen (0.4102), which is justifiable, since, during the composting process, the organic nitrogen within the primary organic materials was degraded into three parts: the first part was mineralized into nitrates and ammonium, from which a part was reincorporated within the metabolism of the active decomposing microorganisms [10], and this was clearly seen in all 9 piles, where significant growth was noted in the microbial count between T0 and T6, and was, therefore, negatively correlated with the C:N ratio (Figure 2). The second part of the organic nitrogen was incorporated within the stabilized organic material of the compost during the humification and the last part was kept free within the compost matrices in the form of mineral nitrogen ($NO_3^-$) [13]. At the end of the composting process, the mineralization process predominated, especially through ammonification [66], and an increase in the total

nitrogen concentration within the residual dry matter was noted. This increase in the total nitrogen in all the piles from T0 until T6 (Figure 3) led to a negative correlation with the C:Ns ($-0.8252$) (Table 3).

**Table 3.** Pearson correlation of all the parameters analyzed during the composting process in the 9 piles.

| Variables | DM | VM | OM | C | pH | EC | P$_2$O$_5$ | K$_2$O | N | C/N | CC | NDF | ADF | ADL |
|---|---|---|---|---|---|---|---|---|---|---|---|---|---|---|
| DM | **1** | | | | | | | | | | | | | |
| VM | $-0.0385$ | **1** | | | | | | | | | | | | |
| OM | **$-0.5055$** | **0.6701** | **1** | | | | | | | | | | | |
| C | **$-0.5057$** | **0.6700** | 1.0000 | **1** | | | | | | | | | | |
| pH | 0.2879 | $-0.0644$ | $-0.0654$ | $-0.0654$ | **1** | | | | | | | | | |
| EC | 0.2431 | **0.4758** | 0.2299 | 0.2297 | $-0.2771$ | **1** | | | | | | | | |
| P$_2$O$_5$ | $-0.0944$ | **0.4797** | 0.4427 | 0.4427 | 0.2216 | 0.0254 | **1** | | | | | | | |
| K$_2$O | **$-0.5090$** | 0.3721 | 0.5519 | 0.5519 | $-0.0002$ | $-0.1008$ | 0.4627 | **1** | | | | | | |
| N | 0.2702 | $-0.2460$ | $-0.1990$ | $-0.1989$ | 0.2713 | 0.1045 | $-0.2923$ | $-0.3278$ | **1** | | | | | |
| C/N | **$-0.4255$** | **0.5837** | **0.6839** | **0.6838** | $-0.2134$ | 0.0648 | 0.4925 | 0.4792 | **$-0.8252$** | **1** | | | | |
| CC | 0.3214 | $-0.1463$ | $-0.2798$ | $-0.2800$ | 0.0939 | 0.0967 | $-0.1787$ | $-0.3541$ | **0.4102** | **$-0.4306$** | **1** | | | |
| NDF | 0.0727 | $-0.0713$ | $-0.1416$ | $-0.1416$ | $-0.0551$ | 0.0874 | $-0.1185$ | $-0.1004$ | $-0.1650$ | 0.0261 | 0.0651 | **1** | | |
| ADF | 0.1422 | $-0.2062$ | $-0.2558$ | $-0.2559$ | $-0.0072$ | 0.1198 | $-0.3737$ | $-0.2394$ | 0.0209 | $-0.1895$ | 0.0530 | **0.6410** | **1** | |
| ADL | 0.1613 | $-0.4549$ | $-0.3539$ | $-0.3538$ | $-0.0083$ | $-0.0960$ | $-0.3557$ | $-0.2116$ | 0.2023 | $-0.4076$ | 0.0678 | 0.3596 | **0.4669** | **1** |

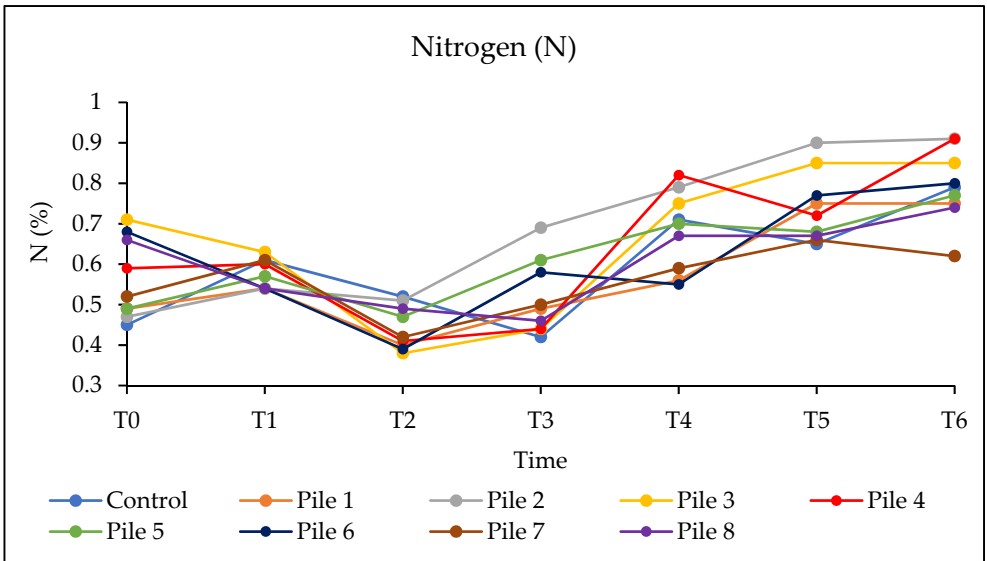

**Figure 3.** Nitrogen variation within the 9 compost piles along the composting process.

At T0, the C:N values were significantly varied from one pile to another and the lowest ratio was found in pile 7 ($31.820 \pm 0.024$). The piles that received a chemical treatment, causing bounds breakage and the loss of free carbon with the rinsing water, were where the C:N ratio was low (Table 4). For the same date, the CC showed a significant variance within the different piles and the highest counts were noted within pile 8 ($1.34 \times 10^6 \pm 2.74 \times 10^3$), with the exception of piles 3 and 4, where the difference was not significant (Table 4). A significant difference was noted between the different piles regarding both the CC and C:N ratio during T2 and T5, with the highest values at T2 for the CC within pile 5 ($1.71 \times 10^6 \pm 2.69 \times 10^3$) and at T5 within pile 2 ($8.51 \times 10^{12} \pm 1.06 \times 10^9$). Regarding the C:N ratio, the lowest value was recorded at T2 for pile 2 ($29.027 \pm 0.025$) and at T5 for pile 2 ($15.147 \pm 0.041$) (Table 4). At T3, a significant variance was noted between the different piles regarding the C:N ratio, with the exception of piles 3 and the control, with the lowest recorded value being noted within pile 2 ($20.417 \pm 0.061$). In addition, a significant variance was noted between the different piles regarding the CC, with the exception of piles 4 and 6, and the highest recorded count was noted within pile 5 ($1.65 \times 10^6 \pm 6.60 \times 10^2$) (Table 4). At T4, a significant variance

was noted between the different piles regarding the CC, with the highest values being within the control ($4.73 \times 10^{11} \pm 4.05 \times 10^8$). For the same date, the C:N ratio showed significant variance within the different piles, with the exception of piles 5 and 8, and the lowest recorded value was noted within pile 2 ($17.300 \pm 0.049$) (Table 4). During the last sampling date, T6, a significant variance was noted between the different piles regarding the C:N ratio, with the exception of piles 4 and 5, and the lowest recorded value was noted within pile 2 ($14.033 \pm 0.020$). For the same date, a significant variance was noted between the different piles regarding the CC, with the lowest noted count being within pile 2 ($1.07 \times 10^{14} \pm 2.37 \times 10^{12}$) (Table 4).

The Volatile Matter (VM), Organic Matter (OM), Organic Carbon (C), and C:N were strongly correlated with each other (*R* between 0.6839 and 0.5837) and, to a lesser extent, with the EC and $P_2O_5$ (*R* between 0.4758 and 0.4797) (Table 3). VM represents the emissions of different gases produced during the composting process, and their patterns showed a decrease with time, especially at thermophilic temperatures (Figure 4a), and this is supported by what was stated in [64,67]. As mentioned previously, the OM and organic carbon were positively correlated; thus, the same trend was observed for both parameters (Figure 4b,c). The lowest decreases in the OM and C were by 12 and 16%, respectively, in pile 1, which was observed from T0 until T6, and the highest decrease was by 47% for the OM and C and was shown in pile 8 (Figure 4b,c), which is in agreement with the literature [68]. The decrease in the OM was due to the mineralization process and compost duration [8], while the decrease in the C was related to its use by the microorganisms for their metabolism, and to the anaerobic reactions that might have taken place in the centers of the piles, causing the emission of methane during the thermophilic phase or volatile organic compounds during the mesophilic phase [69].

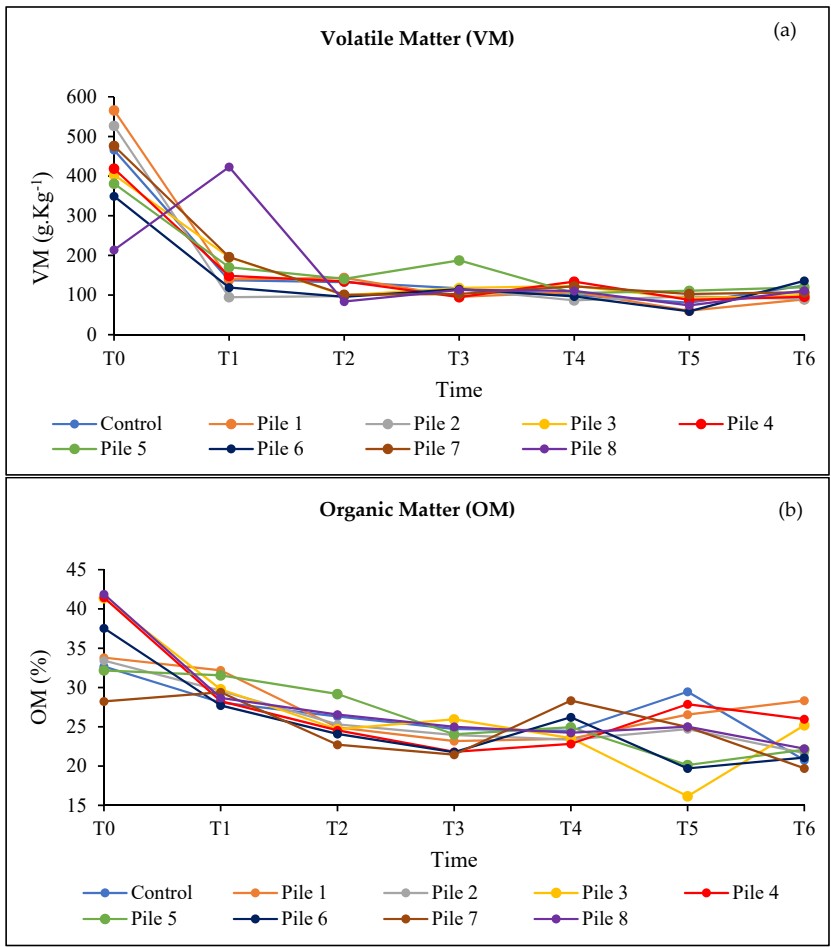

**Figure 4.** *Cont.*

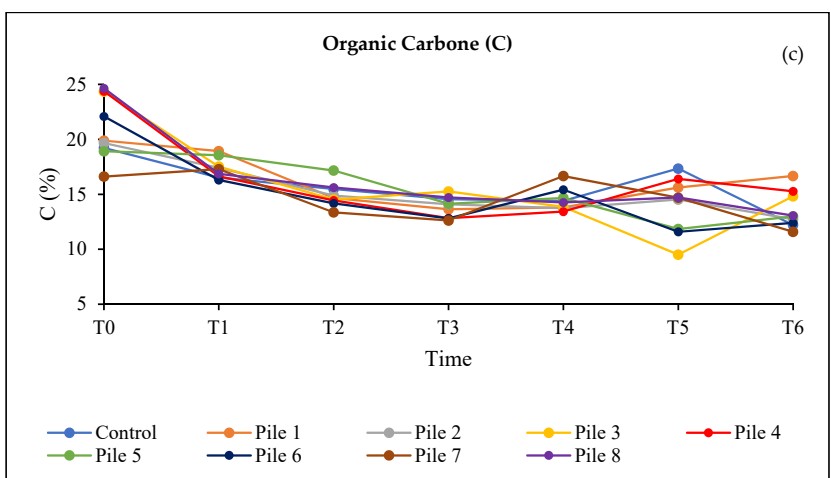

**Figure 4.** Variation of: (**a**) volatile matter, (**b**) organic matter, and (**c**) organic carbon in the piles.

At T0, the Electrical Conductivity (EC) within the different piles ranged between 0.8 and 1.8 mS.cm$^{-1}$, which indicated the low mineral content within the primary used raw materials (Figure 5).

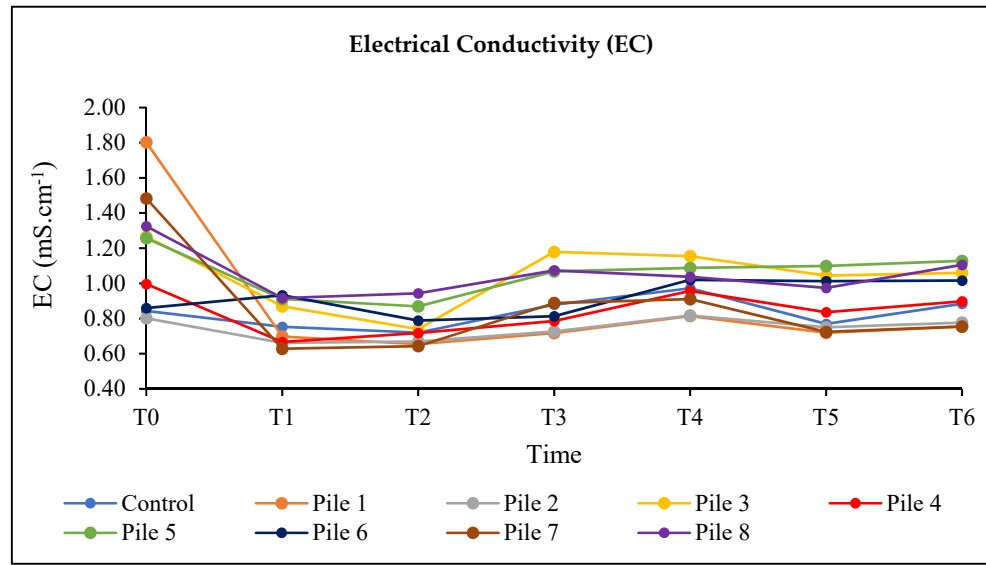

**Figure 5.** Electrical conductivity variation within the 9 compost piles along the composting process.

After T4, a decrease in the EC in all the piles was noted, due to the washing of produced salts from the decomposition process during humidification, or because of their electrical fixation on the stabilized organic matter [70]. The lowest value of EC was observed in pile 7, which reached 0.64 mS.cm$^{-1}$ during the last sampling (Figure 5). The P$_2$O$_5$ content within the nine piles decreased progressively along the composting process, with final values between 0.4 (pile 8) and 0.53% (pile 1), which is in agreement with the literature [71]. The dry matter (DM) was negatively correlated with the OM, C, K$_2$O, and C:N (R between −0.4255 and −0.5090) (Table 3). The DM showed a decrease along the composting process (Figure 6) due to the degradation of the primary unstable organic matter and its concentration into a more stable humic acid form [72].

**Table 4.** Variation of C:N ratio and CC in the different piles.

| Piles | T0 | T1 | T2 | T3 | T4 | T5 | T6 |
|---|---|---|---|---|---|---|---|
| Control | 43.133 ± 0.017 [d] | 26.930 ± 0.049 [d] | 29.833 ± 0.061 [d] | 34.970 ± 0.016 [c] | 20.323 ± 0.053 [d] | 26.770 ± 0.033 [d] | 16.500 ± 0.049 [d] |
| P1 | 40.533 ± 0.233 [a] | 35.150 ± 0.033 [a] | 36.750 ± 0.008 [a] | 27.600 ± 0.163 [a] | 24.543 ± 0.029 [a] | 20.750 ± 0.024 [a] | 22.300 ± 0.049 [a] |
| P2 | 42.013 ± 0.005 [b] | 32.147 ± 0.033 [b] | 29.027 ± 0.025 [b] | 20.417 ± 0.061 [b] | 17.300 ± 0.049 [b] | 15.147 ± 0.041 [b] | 14.033 ± 0.020 [b] |
| P3 | 34.233 ± 0.069 [c] | 27.743 ± 0.028 [c] | 37.813 ± 0.061 [c] | 35.017 ± 0.017 [c] | 19.550 ± 0.033 [c] | 18.250 ± 0.024 [c] | 17.540 ± 0.024 [c] |
| P4 | 41.040 ± 0.024 [e] | 27.580 ± 0.073 [c] | 35.017 ± 0.017 [e] | 29.433 ± 0.037 [d] | 20.350 ± 0.016 [e] | 22.940 ± 0.041 [e] | 16.750 ± 0.033 [e] |
| P5 | 38.583 ± 0.037 [f] | 32.823 ± 0.061 [e] | 36.310 ± 0.057 [f] | 23.250 ± 0.033 [e] | 21.100 ± 0.033 [f] | 17.480 ± 0.033 [f] | 16.823 ± 0.029 [e] |
| P6 | 32.297 ± 0.033 [g] | 30.100 ± 0.065 [f] | 36.063 ± 0.020 [g] | 22.153 ± 0.045 [f] | 27.973 ± 0.012 [g] | 17.050 ± 0.033 [g] | 15.450 ± 0.049 [d] |
| P7 | 31.820 ± 0.024 [h] | 28.530 ± 0.082 [g] | 31.820 ± 0.033 [h] | 25.137 ± 0.041 [g] | 28.163 ± 0.037 [h] | 22.170 ± 0.041 [h] | 18.633 ± 0.045 [f] |
| P8 | 37.330 ± 0.057 [i] | 31.417 ± 0.057 [h] | 32.047 ± 0.025 [i] | 31.833 ± 0.037 [h] | 21.193 ± 0.053 [f] | 21.960 ± 0.024 [i] | 17.710 ± 0.057 [g] |
| | | | | CC | | | |
| Control | $3.57 \times 10^5 \pm 2.69 \times 10^3$ [d] | $8.17 \times 10^5 \pm 2.29 \times 10^3$ [d] | $1.39 \times 10^6 \pm 1.51 \times 10^3$ [d] | $1.64 \times 10^6 \pm 2.74 \times 10^3$ [d] | $4.73 \times 10^{11} \pm 4.05 \times 10^8$ [d] | $6.80 \times 10^{12} \pm 2.74 \times 10^9$ [d] | $2.31 \times 10^{14} \pm 9.42 \times 10^{11}$ [d] |
| P1 | $3.43 \times 10^5 \pm 2.74 \times 10^3$ [a] | $1.97 \times 10^6 \pm 2.74 \times 10^3$ [a] | $8.63 \times 10^5 \pm 2.74 \times 10^3$ [a] | $5.43 \times 10^5 \pm 2.74 \times 10^3$ [a] | $4.05 \times 10^{11} \pm 4.92 \times 10^7$ [a] | $6.48 \times 10^{12} \pm 1.28 \times 10^9$ [a] | $1.51 \times 10^{14} \pm 7.85 \times 10^{11}$ [a] |
| P2 | $1.63 \times 10^5 \pm 2.74 \times 10^3$ [b] | $1.90 \times 10^5 \pm 8.16 \times 10^1$ [b] | $8.81 \times 10^5 \pm 8.16 \times 10^2$ [b] | $3.50 \times 10^5 \pm 3.40 \times 10^2$ [b] | $3.51 \times 10^{11} \pm 2.74 \times 10^8$ [b] | $8.51 \times 10^{12} \pm 1.06 \times 10^9$ [b] | $1.07 \times 10^{14} \pm 2.37 \times 10^{12}$ [b] |
| P3 | $1.03 \times 10^5 \pm 2.74 \times 10^3$ [c] | $2.70 \times 10^5 \pm 1.25 \times 10^2$ [c] | $5.41 \times 10^5 \pm 7.13 \times 10^2$ [c] | $5.07 \times 10^5 \pm 1.51 \times 10^3$ [c] | $4.34 \times 10^{11} \pm 4.5 \times 10^8$ [c] | $7.09 \times 10^{12} \pm 2.74 \times 10^9$ [c] | $6.97 \times 10^{14} \pm 2.12 \times 10^{12}$ [c] |
| P4 | $1.01 \times 10^5 \pm 8.16 \times 10^3$ [c] | $8.07 \times 10^5 \pm 1.88 \times 10^3$ [e] | $7.61 \times 10^5 \pm 4.55 \times 10^2$ [e] | $7.27 \times 10^5 \pm 1.39 \times 10^3$ [e] | $4.63 \times 10^{11} \pm 4.05 \times 10^8$ [e] | $6.90 \times 10^{12} \pm 4.08 \times 10^8$ [e] | $3.00 \times 10^{14} \pm 3.86 \times 10^{11}$ [e] |
| P5 | $1.33 \times 10^5 \pm 8.16 \times 10^3$ [e] | $2.60 \times 10^6 \pm 5.25 \times 10^2$ [f] | $1.71 \times 10^6 \pm 2.69 \times 10^3$ [f] | $1.65 \times 10^6 \pm 6.60 \times 10^2$ [f] | $3.95 \times 10^{11} \pm 2.37 \times 10^8$ [f] | $5.88 \times 10^{12} \pm 6.68 \times 10^8$ [f] | $5.27 \times 10^{14} \pm 2.69 \times 10^{12}$ [f] |
| P6 | $1.01 \times 10^6 \pm 2.45 \times 10^3$ [f] | $7.47 \times 10^5 \pm 1.59 \times 10^3$ [g] | $4.77 \times 10^5 \pm 5.72 \times 10^2$ [g] | $7.31 \times 10^5 \pm 4.11 \times 10^2$ [e] | $4.48 \times 10^{11} \pm 4.32 \times 10^7$ [g] | $6.45 \times 10^{12} \pm 2.74 \times 10^9$ [g] | $7.83 \times 10^{14} \pm 2.74 \times 10^{12}$ [g] |
| P7 | $6.70 \times 10^5 \pm 8.16 \times 10^3$ [g] | $6.07 \times 10^5 \pm 1.14 \times 10^3$ [h] | $1.06 \times 10^6 \pm 1.39 \times 10^3$ [h] | $7.47 \times 10^5 \pm 1.51 \times 10^3$ [g] | $4.56 \times 10^{11} \pm 2.45 \times 10^8$ [h] | $5.46 \times 10^{12} \pm 8.38 \times 10^8$ [h] | $6.43 \times 10^{14} \pm 2.74 \times 10^{12}$ [h] |
| P8 | $1.34 \times 10^6 \pm 2.74 \times 10^3$ [h] | $3.87 \times 10^5 \pm 2.37 \times 10^3$ [i] | $1.50 \times 10^6 \pm 8.29 \times 10^2$ [i] | $1.22 \times 10^6 \pm 5.10 \times 10^2$ [h] | $3.37 \times 10^{11} \pm 2.74 \times 10^8$ [i] | $6.56 \times 10^{12} \pm 2.74 \times 10^9$ [i] | $4.87 \times 10^{14} \pm 7.35 \times 10^{11}$ [i] |

Note: The averages within columns followed by the same letter do not significantly differ according to Tukey's multiple range test (α = 0.05). T0: Sampling at day 0, T1: Sampling at day 15, T2: Sampling at day 30, T3: Sampling at day 45, T4: Sampling at day 60, T5: Sampling at day 75, and T6: Sampling at day 90.

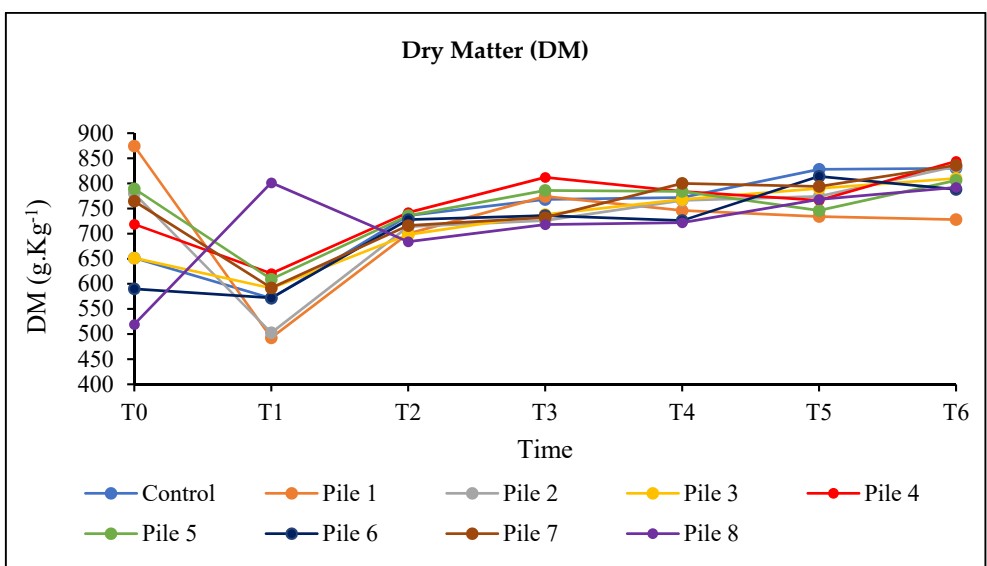

**Figure 6.** Dry matter variation within the 9 compost piles along the composting process.

*3.4. Evaluation of the Quality of the Final Compost*

The final compost quality is always identified as a major problem in determining whether a compost could be used as soil amendment. The most important parameters that must be tackled within the norms, ensuring environmental protection, public health, and the soil, include: microbial pathogens, physico-chemical characteristics, both potentially toxic organic and inorganic compounds (heavy metals, phtalates, and polycyclic aromatic hydrocarbons), and stability [73].

The pH values varied between 8.94 and 9.71 and fell above the recommended limits (Table 5), but were less than the range of pH between 7 and 9 [74]. The C:N ratio was above the standard, except for pile 2, as for the $K_2O$ content, its value was in the range of 0.25%–0.5%, except for piles 1 and 7. The mercury (Hg) ranged between 0.0212 ppm and 0.0697 ppm, which was less than the standards (Table 5). For this, it is recommended to fulfill a detailed profiling of the Lebanese primary organic raw materials used in composting systems, and the same goes for the used water for humidification, which will help to elaborate an adequate Lebanese standard for compost.

**Table 5.** Chemical parameters of the final compost.

| Compost Parameters | Pile Number | | | | | | | | | Standard [75] |
|---|---|---|---|---|---|---|---|---|---|---|
| | Control | 1 | 2 | 3 | 4 | 5 | 6 | 7 | 8 | |
| pH | 9.21 | 9.95 | 9.71 | 9.15 | 9.31 | 9.12 | 8.94 | 9.8 | 9.11 | 7–8 |
| EC mS.cm$^{-1}$ | 0.8841 | 0.7545 | 0.7769 | 1.06 | 0.8981 | 1.128 | 1.016 | 0.7547 | 1.105 | <4 |
| O.M. (%) | 23.7 | 36.12 | 25.62 | 29 | 28.02 | 31.38 | 34.8 | 27.1 | 31.92 | - |
| N (%) | 0.64 | 0.63 | 0.79 | 0.73 | 0.67 | 0.78 | 0.82 | 0.6 | 0.75 | <2 |
| C:N | 21.78 | 33.73 | 19.07 | 23.37 | 24.59 | 23.66 | 24.96 | 26.56 | 25.02 | <20 |
| $K_2O$ (%) | 0.484 | 0.639 | 0.359 | 0.387 | 0.399 | 0.415 | 0.452 | 0.545 | 0.358 | 0.25–0.5 |
| $P_2O_5$ (%) | 0.4 | 0.53 | 0.49 | 0.44 | 0.52 | 0.4 | 0.45 | 0.49 | 0.4 | 0.7–0.9 |
| Hg (ppm) | 0.0538 | 0.0633 | 0.0677 | 0.0656 | 0.0510 | 0.0227 | 0.0697 | 0.0212 | 0.0517 | ≤8 ppm |

Regarding the microbiological profiling, the manure used in compost piles usually contains several pathogens for humans, the concentrations of which differ from one animal species to another [76]. The microbial contents in the nine piles at T6 are shown in Table 6. *Salmonella* sp. was absent in all the piles, *Enterococci* sp. varied between $10^2$ CFU/g in pile 2 and $4 \times 10^3$ CFU/g in the control pile, and *Staphylococcus aureus* varied between $2 \times 10^2$ CFU/g in pile 2 and $6 \times 10^3$ CFU/g in the control, which was below the microbio-

logical limits. It was noticed in the control pile that the concentrations of *Escherichia coli* and *Bacillus* sp. were above the admissible limits set by the EPA (2008), with respective cell concentrations of $2 \times 10^3$ and $2 \times 10^5$ CFU/g. The *Bacillus* sp was also higher than the admissible limit in piles 7 and 8, with a value of $10^5$ (Table 6).

**Table 6.** Microbiological profiling of the final compost in the 9 piles.

| Detectable Microorganisms | *Salmonella* sp. | *Staphylococcus aureus* | *Enterococci* spp. | *Escherichia coli* | *Bacillus* spp. |
|---|---|---|---|---|---|
| Microbiological limits (EPA 2008) | Absence in 25 g | $10^2$–$10^4$ CFU/g | 5000 CFU/g | 1000 CFU/g | $10^3$–$10^5$ CFU/g |
| Control | Absence | $6 \times 10^3$ | $4 \times 10^3$ | $2 \times 10^3$ ** | $2 \times 10^5$ ** |
| Pile 1 | Absence | $10^3$ | $10^3$ | $3 \times 10^2$ | $10^4$ |
| Pile 2 | Absence | $2 \times 10^2$ | $10^2$ | $10^2$ | $10^3$ |
| Pile 3 | Absence | $2 \times 10^3$ | $10^2$ | $2 \times 10^2$ | $10^4$ |
| Pile 4 | Absence | $4 \times 10^3$ | $10^3$ | $5 \times 10^2$ | $2 \times 10^4$ |
| Pile 5 | Absence | $2 \times 10^3$ | $10^3$ | $2 \times 10^2$ | $2 \times 10^3$ |
| Pile 6 | Absence | $3 \times 10^3$ | $2 \times 10^3$ | $6 \times 10^2$ | $2 \times 10^4$ |
| Pile 7 | Absence | $5 \times 10^2$ | $3 \times 10^3$ | $2 \times 10^2$ | $10^5$ ** |
| Pile 8 | Absence | $4 \times 10^3$ | $2 \times 10^3$ | $9 \times 10^2$ | $10^5$ ** |

** exceed the acceptable microbiological level.

It is clearly seen that the application of a higher concentration (2.5%) of the adapted microbial inoculum within pile 2, with an antagonistic ability toward some of the tested pathogens (Table 2), gave a better sanitation efficiency in the final compost than the control and piles 3 and 8.

## 4. Conclusions

The combined pretreatment of the lignocellulosic fraction with NaOH (10%) and a developed, adapted bacterial inoculum from compost piles under Mediterranean conditions, allowed for the launching, maintenance, and quickening the composting process by 15 days. This combination eliminated the inhibitory action of lignin on the existing microbiota, allowing for an easier hydrolysis of the cellulose and hemicelluloses. This 'tailor-made' process was meticulously planned and selected based on the characteristic properties of Mediterranean biomass. The findings of this study show the importance of both biomass profiling and microbiota selection within the same geographical zones for the best compost mix set and the ideal interactions and decomposition.

**Author Contributions:** Conceptualization, A.T., R.G.M. and Z.H.; methodology, A.T., D.S. and Z.R.; formal analysis, A.T.; investigation, A.T., D.S. and Z.R.; resources, A.T.; data curation, C.G. and M.D.; writing—original draft preparation, A.T. and V.A.; writing—review and editing, A.T., Z.R., D.S., R.G.M., V.A. and Z.H.; supervision, D.S. and R.G.M. All authors have read and agreed to the published version of the manuscript.

**Funding:** This research received no external funding.

**Data Availability Statement:** Data are contained within the article.

**Acknowledgments:** The financial support of the study from the Lebanese Agricultural Research Institute (LARI), and Saint Joseph University Faculty of Sciences are gratefully acknowledged.

**Conflicts of Interest:** The authors declare no conflict of interest.

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
