# Peer review of "Study of Raw Material Pretreatment and the Microbiota Selection Effect on the Composting Process Efficiency"

_agronomy, doi:10.3390/agronomy13082048_

Round 1
Reviewer 1 Report
Dear Authors, here are my issues and comments regarding your paper:
- the term hemicelluloses exists only as plural noun;
- lines 122-123 should be rewritten...
- chemical composition of manure and the rest of biomaterials must be detailed;
- what happens to the wood remnants? are they completelly metabolised? if not what is the chemical composition of the remaining parts?
- the procedure regarding determination of the hemicelluloses content should be briefly described in terms of procedural steps and also chemicals used.
- what happens to the other wood components?
- in table 3 the variation reason of C:N rati in the series 43.13; 26.93; 29.83; 34,97; 20.32; 26.77; 16.50 seems strange. Is there any reason?
hemicelluloses is a noun which supports only plural.
-minor spelling mistakes were detected such as the obtained ..line 352 spelling and grammar check is highly recommended.
Author Response
Dear Authors, here are my issues and comments regarding your paper:
Thank you for your time in reviewing our manuscript.
- The term hemicelluloses exists only as plural noun
Done
- Lines 122-123 should be rewritten...
Done
- chemical composition of manure and the rest of biomaterials must be detailed
Table added showing covered parameters within Supplementary Information
- What happens to the wood remnants? Are they completely metabolized? If not what is the chemical composition of the remaining parts?
Wood remnants degradation process were followed during the entire composting time represented in our study, evaluating the quantities of lignin, cellulose and hemicelluloses. The final remaining quantities within every treatment (compost piles) are represented in the sampling number T6 (after 90 days). At field level, within a composting facility these non-decomposed fractions are eliminated through the trammel (Ø > 0.5 cm) and re-introduced to newly established compost piles again.
- The procedure regarding determination of the hemicelluloses content should be briefly described in terms of procedural steps and also chemicals used.
Done
- What happens to the other wood components?
Being shredded to the residual size of 3 to 5 cm allows the microbiota to degrade them properly along the duration of 90 days. As explained within comment 4; at field level, within a composting facility these non-decomposed fractions are eliminated through the trammel (Ø > 0.5 cm) and re-introduced to newly established compost piles again to be subjected to additional degradation by existing microbiota.
- In table 3 the variation reason of C:N ratio in the series 43.13; 26.93; 29.83; 34,97; 20.32; 26.77; 16.50 seems strange. Is there any reason?
Composting is an aerobic degradation process witnessing complex biochemical processes affected by several factors including the microbial activity, climatic conditions, and humidity of the organic material, type and residual size of the used primary organic raw material. Facing these facts these variations are reasonable, along the phases of composting which were relatively stabilized during the curing phase at T6 were most of these processes become less active.
Comments on the Quality of English Language
Hemicelluloses is a noun which supports only plural.
Done
Minor spelling mistakes were detected such as the obtained line 352 spelling and grammar check is highly recommended.
Reviewed and checked as requested
Thank you for your time in reviewing our manuscript.
Reviewer 2 Report
Comments
Line 24 to 27 is not clear.
Line 36 what is the effect on composting efficiency of each temperature range?
Line 41 you should describe the different step of a compost. Hydrolysis step, follow often by acidogenese etc.
Line 114 where the furfural is generated? From the degradation of hemicellulose or lignin. Please bring clarity in the discussion.
Line 132 Among the pretreatment method, extrusion or reactive extrusion is getting more attention. It should be among the pretreatment that was discussed.
Line 133 the biological process can be environmentally friendly, but the process may be slow. You should cover the advantages and drawbacks of the technology discussed.
Line 144 where is this study mentioned as previous work? Any references?
Line 159 most of the existing process are doing this process without robust inoculum and without removing existing impurities because the process is working just fine, and they are saving cost for the process. This should be mentioned in your discussion. Now your process should be competed with the existing process with efficiency.
Line 173 what was the characteristics of cow manure used: percentage of humididity, nitrogen concentration, total solid, total solid volatile, ashes etc.
Same questions for the mix of wood: lignin, hemicellulose, cellulose percentage of each fraction, humditity etc…
What was the normality or molarity of NaOH and HCl.
What was the composition of 1% inoculum and 2.5% inoculum.
Do you have the name of the strains used in the collection? If yes you should mentioned it.
These informations should be described before the table 1.
Line 293 the pH decreases because of the accumulation of organic compounds such as acetone, butanol, propanol, lactate, butyrate, etc. after the first phase which consists of hydrolysis of complex molecules. The discussion should be correct.
Also, the pH of the control might be too high. It should have been maintained around 7 or at least try to be maintain around a fix value to observe the effect of pH which is with temperature and humidity the main parameters.
Line 467 and 479 showed that the experiments were not above the results of the control. Some study are lacking, as the effect of pH control, humidity percentage, temperature for the two inoculum. The methodology should have focused on these parameters.
english should be improved
Author Response
Thank you for your time in reviewing our manuscript.
- Line 24 to 27 is not clear.
Abstract reformulated and more explanation were given (Lines 14 – Line 33).
- Line 36 what is the effect on composting efficiency of each temperature range?
It differs in the responsible active microbiota fulfilling the decomposition of the existing organic materials within the compost piles as follow:
In the first mesophile phase (first 24 – 48 hours) organisms (gathering fungi and mesophile bacteria) use the readily degradable compounds such as simple sugars; in the second thermophile phase organisms especially bacillus and thermus breakdown complex polymers and most of soil insects, plant and human pathogens are destroyed, coupled with the release of phytotoxins and ammonia are observed, and in the final second mesophile phase fungi are active again and the humic fraction is stabilised (Chen et al., 1996)
- Line 41 you should describe the different step of a compost. Hydrolysis step, follow often by acidogenese etc.
Not applicable because in our study we adopted composting using windrow method which is an aerobic process for the degradation of organic materials. Listed steps applied under anaerobic fermentation processes.
- Line 114 where the furfural is generated? From the degradation of hemicellulose or lignin. Please bring clarity in the discussion.
During acidic pretreatment, the H+ plays a role in hydrolyzing both polysaccharides and lignin, resulting in a net disruption of the tightly woven lignocellulosic matrix. Physicochemical changes of biomass, such as depolymerization, morphological changes, as well as many additional structural changes take place to varying degrees depending on pretreatment severity (Donohoe et al., 2011; Kumar et al., 2009). As a result, the pretreatment of biomass generates a liquid and solid phase that contains lignin fragments and process able saccharides (i.e., poly-, oligo-, and mono-) amongst other decomposition products (Huang et al., 2019).
Along with the cellulose-enriched solid phase generated from pretreatment, several fermentable monosaccharides and their degradation compounds, including furfural (FF), 5-hydroxymethylfurfural (5-HMF), levulinic acid and formic acid are found in the liquid phase.
- Line 132 Among the pre-treatment method, extrusion or reactive extrusion is getting more attention. It should be among the pre-treatment that was discussed.
Extrusion technology highlighted and reference added
- Line 133 the biological process can be environmentally friendly, but the process may be slow. You should cover the advantages and drawbacks of the technology discussed.
Done. And added to the manuscript
- Line 144 where is this study mentioned as previous work? Any references?
The work was done and patented (reference 40). The reference [36] Tannoury et al. 2022 was added to the manuscript.
- Line 159 most of the existing process are doing this process without robust inoculum and without removing existing impurities because the process is working just fine, and they are saving cost for the process. This should be mentioned in your discussion. Now your process should be competed with the existing process with efficiency.
We didn’t discuss this issue because we didn’t use comingled wastes. In our experiment, the clean organic raw material (animal manure and shredded wood from agriculture) was used in our study, and the different treatments were compared to a control pile and results were compared accordingly for each parameter.
- Line 173 what was the characteristics of cow manure used: percentage of humidity, nitrogen concentration, total solid, total solid volatile, ashes etc. Same questions for the mix of wood: lignin, hemicellulose, cellulose percentage of each fraction, humditity etc…
Done added under Supplementary information (Table S1)
- What was the normality or molarity of NaOH and HCl.
NaOH is 0.1 M and HCl is 3.2 M. These values were added to the manuscript
- What was the composition of 1% inoculum and 2.5% inoculum.do you have the name of the strains used in the collection? If yes you should mentioned it. These information’s should be described before the table 1.
Developed in the reference 36 and added to the manuscript before the table 1 as suggested.
- Line 293 the pH decreases because of the accumulation of organic compounds such as acetone, butanol, propanol, lactate, butyrate, etc. after the first phase which consists of hydrolysis of complex molecules. The discussion should be correct.
This status is signaled in the following lines and were also developed to fit your recommendations
- Also, the pH of the control might be too high. It should have been maintained around 7 or at least try to be maintain around a fix value to observe the effect of pH which is with temperature and humidity the main parameters.
To note that other than the different pre-treatment of the primary organic raw materials by chemical /microbiological alone or mixed were not fulfilled within the field trials. Only the temperature and humidity were monitored and no buffering for the pH was fulfilled for any of the treatments along the entire experiments, that’s why the values were relatively high
- Line 467 and 479 showed that the experiments were not above the results of the control. Some study are lacking, as the effect of pH control, humidity percentage, temperature for the two inoculum. The methodology should have focused on these parameters.
To note that other than the different pre-treatment of the primary organic raw materials by chemical, microbiological alone or mixed were not fulfilled within the field trials (no buffering for the pH ….), and variable results were obtained for the different parameters showing advantages/disadvantages in comparison to the control and among each other. The affinity for the best treatment gaining higher advantages regarding the different parameters tested within this study was confirmed through the Pearson correlation analysis fulfilled to the totality of these parameters.
Thank you for your time in reviewing our manuscript.
Reviewer 3 Report
The article "Study of raw material pretreatment and microbiota selection effect on the composting process efficiency" is interesting and well written. In this study they looked at 9 different compost piles subjected to chemical and microbiological treatments. They found that a specific microbial inoculum at 2.5% concentration with 10% NaOH ensured a robust and faster composting process by 15 days. The introduction was well written by reviewing the relevant literature. Methods were also well described. Results were presented in a good format by visualising the data and providing a nice discussion along with the results. However, there are few minor suggestions to improve the article, so I recommend the article for publication after a minor revision.
1. Lines 45-47: Please cite a reference article.
2. Lines 144-155: Please cite an article if it has already been published.
3. Why did you choose HCl 10% and NaOH 10% for this study, any particular reason, please explain?
4. Also, why did you choose these particular concentrations, inoculum 1% and inoculum 2.5% for this study?
5. Lines 244-246: What is the optimum pile size (in kg) to achieve these values?
6. Figure 1: instead of mentioning time as T0, T1, T2...., please mention it in hours, it will be easier for the readers to follow.
7. Lines 288- 355: Please also mention the time in hours instead of T0, T1, T2... (as mentioned in the previous section).
8. Figure 2: Please also expand the ADF, NDf, ADL... in the figure legend, it will be easy to follow for the readers.
9. Similarly, please indicate the time in hours in Figures 3 - 6 and Table 3.
10. Figure 3: Apply these error values (standard deviation) in the figure, it would be clear.
English language of the article is fine, only mior editing is required.
Author Response
The article "Study of raw material pretreatment and microbiota selection effect on the composting process efficiency" is interesting and well written. In this study they looked at 9 different compost piles subjected to chemical and microbiological treatments. They found that a specific microbial inoculum at 2.5% concentration with 10% NaOH ensured a robust and faster composting process by 15 days. The introduction was well written by reviewing the relevant literature. Methods were also well described. Results were presented in a good format by visualizing the data and providing a nice discussion along with the results. However, there are few minor suggestions to improve the article, so I recommend the article for publication after a minor revision.
Thank you for your time in reviewing our manuscript.
- Lines 45-47: Please cite a reference article.
Done reference (4) was added to the manuscript (
- Lines 144-155: Please cite an article if it has already been published.
Done reference (36) was added to the manuscript
- Why did you choose HCl 10% and NaOH 10% for this study, any particular reason, please explain?
They have been used in different works by previous colleagues (Achkar, 2017) and within the bibliography with positive results, and they were available in our stock
- Also, why did you choose these particular concentrations, inoculum 1% and inoculum 2.5% for this study?
Each selected bacterial strains was grown apart to reach a cell concentration of 1 × 108 CFU.mL−1 as stated by (Zhao et al., 2016). We started with a first set of trials with 1% inoculum which didn’t show any significant variation. Thus, we increased the concentration of applied inoculum to 2.5%
- Lines 244-246: What is the optimum pile size (in kg) to achieve these values?
In our recent work (unpublished yet) field trials with compost piles of 2000 kg weight were validated even under harsh conditions (in an area of 1500m altitude from sea level and night temperatures falling below 5 °C), and treated with our microbial inoculums; compost piles reached a temperature of 70°C
- Figure 1: instead of mentioning time as T0, T1, T2...., please mention it in hours, it will be easier for the readers to follow.
Thank you for your advice but we will keep it as is because it doesn’t fit on the figures, and additional explanation were given within materials and methods for the duration of each sampling in days and hours
- Lines 288- 355: Please also mention the time in hours instead of T0, T1, T2... (as mentioned in the previous section).
We will keep it as is because we couldn’t change it in the figures nor the tables. To note that additional explanation were given within materials and methods for the duration of each sampling in days and hours
8- Figure 2: Please also expand the ADF, NDf, ADL... in the figure legend, it will be easy to follow for the readers. Done
9- Similarly, please indicate the time in hours in Figures 3 - 6 and Table 3. Thank you for your advice but we will keep it as is because it doesn’t fit on the figures. To note that additional explanation were given within materials and methods for the duration of each sampling in days and hours
10- Figure 3: Apply these error values (standard deviation) in the figure, it would be clear.
We tried to add it but the figure is not clear
Comments on the Quality of English Language
English language of the article is fine, only minor editing is required.
Thank you for your time in reviewing our manuscript.
Reviewer 4 Report
The manuscript presents study of raw material pretreatment and microbiota selection effect on the composting process efficiency. Overall, the topic is interesting, and the manuscript got some reliable data from experiments. The presented manuscript is written reasonably and the data is clear. I recommend it be published after minor revisions.
1. Abstract: Reduce the description of the background and extend the results of the paper.
2. Introduction. Reduce unnecessary segments and rearrange the logical relationship of this part.
3. Materials and Methods Line 177, authors said ‘……, containing 70% manure (90 Kg) and 40% wood remnants (40 Kg)……’. The piles are up to more than 100%? please double check the data here.
4. Line 97-98, Sampling was conducted seven times; T0 represents the launching date of the compost piles, T1 after 15 days of composting, T2 to T6 represent 30, 45, 60, 75 and 90 days of the compost piles construction. In my opinion, use days to indicate the composting time directly is much better than current presentation. As shown in Figure 1 and Figure 2, the use of composting days is much better than T0, T1……
5. The second sub-figure caption of Figure 1 is incomplete. I mean pile 8 just see pile.
6. Title 3.3 is incomplete. Principal Component data Analysis of …………
Author Response
The manuscript presents study of raw material pretreatment and microbiota selection effect on the composting process efficiency. Overall, the topic is interesting, and the manuscript got some reliable data from experiments. The presented manuscript is written reasonably and the data is clear. I recommend it be published after minor revisions.
Thank you for your time in reviewing our manuscript.
- Abstract: Reduce the description of the background and extend the results of the paper.
The Abstract was reformulated and more results are shown as requested
- Reduce unnecessary segments and rearrange the logical relationship of this part.
Introduction reviewed
- Materials and Methods Line 177, authors said ‘……, containing 70% manure (90 Kg) and 40% wood remnants (40 Kg)……’. The piles are up to more than 100%? please double check the data here.
Done
- Line 197-198, Sampling was conducted seven times; T0 represents the launching date of the compost piles, T1 after 15 days of composting, T2 to T6 represent 30, 45, 60, 75 and 90 days of the compost piles construction. In my opinion, use days to indicate the composting time directly is much better than current presentation. As shown in Figure 1 and Figure 2, the use of composting days is much better than T0, T1……
Thank you for your suggestion but we prefer to keep it as is because we don’t have time to change it for the entire parameters throughout the manuscript. To note that this section was reformulated and exact timing in days and hours were added for each timing adopted.
- The second sub-figure caption of Figure 1 is incomplete. I mean pile 8 just see pile.
Adjusted as requested
- Title 3.3 is incomplete. Principal Component data Analysis of …………
Adjusted as requested
Thank you for your time in reviewing our manuscript
Round 2
Reviewer 1 Report
Dear authors,
I recommend including in the manuscript the table shown in supplementary at this moment. Moreover the table should reveal (at least) the initial chemical composition of wood remnant and also the final (after composting) in terms of cellulose, lignin and hemicelluloses.
Author Response
I recommend including in the manuscript the table shown in supplementary at this moment. Moreover the table should reveal (at least) the initial chemical composition of wood remnant and also the final (after composting) in terms of cellulose, lignin and hemicelluloses.
The table was included to the manuscript and entitled Table 2 and all tables were adjusted accordingly. Values of Lignin, cellulose and hemicellulose by the end of the composting was added to the table too. No supplementary information do exist for our manuscript.
Reviewer 2 Report
The manuscript was improved and can warrant publication in agronomy.